# Context-Aware Indoor Point Cloud Object Generation through User Instructions

## ABSTRACT

Indoor scene modification has emerged as a prominent area within computer vision, particularly for its applications in Augmented Reality (AR) and Virtual Reality (VR). Traditional methods often rely on pre-existing object databases and predetermined object positions, limiting their flexibility and adaptability to new scenarios. In response to this challenge, we present a novel end-to-end multi-modal deep neural network capable of generating point cloud objects seamlessly integrated with their surroundings, driven by textual instructions. Our model revolutionizes scene modification by enabling the creation of new environments with previously unseen object layouts, eliminating the need for pre-stored CAD models. Leveraging Point-E as our generative model, we introduce innovative techniques such as quantized position prediction and Top-K estimation to address the issue of false negatives resulting from ambiguous language descriptions. Furthermore, we conduct comprehensive evaluations to showcase the diversity of generated objects, the efficacy of textual instructions, and the quantitative metrics, affirming the realism and versatility of our model in generating indoor objects. To provide a holistic assessment, we incorporate visual grounding as an additional metric, ensuring the quality and coherence of the scenes produced by our model. Through these advancements, our approach not only advances the state-of-the-art in indoor scene modification but also lays the foundation for future innovations in immersive computing and digital environment creation. The anonymized project is available at https://anonymous.4open.science/r/Context-aware-Indoor-PCG-9DFB.

## CCS CONCEPTS

• **Computing methodologies → Computer vision**; **Scene understanding**.

## KEYWORDS

Deep Learning, 3D Point Clouds, Generative Model

**ACM Reference Format:**
Anonymous Author(s). 2024. Context-Aware Indoor Point Cloud Object Generation through User Instructions. In *Proceedings of the 32nd ACM International Conference on MultiMedia (MM'24)*. ACM, New York, NY, USA, 9 pages. https://doi.org/XXXXXXX.XXXXXXX

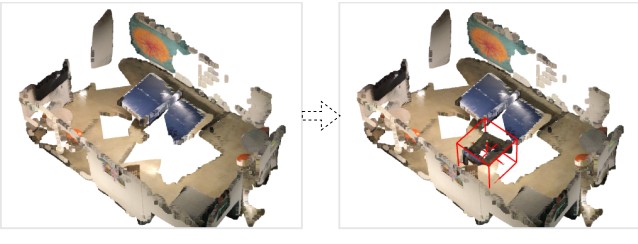

Query: "Generate a couch that is closer to the TV."

**Figure 1: Our model generates a couch that is positioned close to the television in response to the query and makes it consistent with the rest of the scene, i.e., the orientation, size, and overlap with other objects in certain cases.**

## 1 INTRODUCTION

In the rapidly evolving field of computer vision, the significance of 3D computer vision has reached unprecedented heights. It poses many challenges that are similar to those in 2D image processing but also offers the opportunity to leverage successful strategies from the 2D domain on classic tasks such as object detection. However, the complex data structure of point clouds and the nature of 3D scenes present challenges for tasks like modifying 3D scenes. Applying experience from 2D image processing is limited in this context. This paper discusses a new task called scene modification, which aims to modify a point cloud-based 3D scene according to user instructions, and proposes a solution for it.

Scene modification aims to create new scenarios with previously unseen layouts of objects, thereby enriching the geometrical and auxiliary color features according to the will of the user. For instance, as depicted in Fig. 1, given a specific scene and query, an object that harmonizes with its surroundings should be generated and inserted in the correct place by the model.

Scene modification also has a wide range of applications in industries. It plays a crucial role in the fields of Augmented Reality (AR) and Virtual Reality (VR). In AR, it is used to superimpose virtual objects onto the real world, enhancing the user's perception and interaction with their environment [21]. In VR, scene modification is used to create immersive virtual environments. It can generate diverse scenarios by adding or modifying objects in a virtual scene, enriching the user's experience. Nevertheless, in today's VR and AR software development process, it is necessary to have a relatively large material library to insert different types of objects into the scene. Our method, however, allows developers and artists to create realistic and consistent objects directly using simple text prompts, freeing up large amounts of storage and reducing time costs.

In this paper, we focus on 3D scene modification of point clouds, in which point clouds serve as the fundamental building blocks for

creating complex and detailed 3D objects. Previous works mainly focus on the transformation of a single object [20] or generating point clouds from existing ones [23]. Some works on scene modification reply on inserting pre-built CAD objects into scene [28, 33, 34, 38]. While scene modification can be simplified to selection and insertion as a two-stage pipeline, this may result in inflexibility and inconsistency with the surroundings, and only a few works address the issue of generating new objects and incorporating them into scenes as an end-to-end process.

To address these limitations, we propose an end-to-end multimodal deep neural network. It can generate objects consistent with the surroundings and integrate these objects seamlessly into given scenes, conditioned on text instructions. This work introduces a unique data pipeline, empowered by GPT, to transform existing visual grounding datasets to apply them to the task of instructed scene modification (Sec. 3.1). Moreover, a feature fusion module has been designed for space-text feature fusion. After extracting the spatial features from the 3D scene and textual features from the text query, these features will be fused as fusion features. The fusion features, derived from the cross-attention mechanism, capture high-level information from both the surroundings and the query texts, enabling conditional object generation and the location prediction of target point clouds (Sec. 3.2, 3.3 and 3.4).

The effectiveness of our proposed method is validated through qualitative and quantitative experiments conducted on the ReferIt3D dataset [1] (Sec. 4). This work, therefore, presents a significant contribution to this field by addressing previous limitations and proposing innovative solutions.

In summary, the contributions of our work are as follows:

- We generate a new dataset for scene modification tasks by designing a GPT-aided data pipeline for paraphrasing the descriptive texts in ReferIt3D dataset to generative instructions, referred to Nr3D-SA and Sr3D-SA datasets. The dataset will be released to the public and can be utilized for comparable tasks in subsequent studies.
- We propose an end-to-end multi-modal diffusion-based deep neural network model for generating in-door 3D objects into specific scenes according to input instructions.
- We propose quantized position prediction, a simple but effective technique to predict Top-K candidate positions, which mitigates false negative problems arising from the ambiguity of language and provides reasonable options.
- We introduce the visual grounding task as an evaluation strategy to assess the quality of a generated scene and integrate several metrics to evaluate the generated objects.

## 2 RELATED WORK

*Text-guided 3D Vision.* While 2D text-guided tasks have achieved great success in recent years, 3D text-guided tasks also hold a high degree of research interest. The majority of 3D V+L tasks are derived from corresponding 2D tasks as an extension of 2D space to 3D space, such as 3D visual grounding [1, 6, 15, 41], 3D dense captioning [8, 9, 17], and 3D shape generation [7, 22, 39]. Despite the differences between these 3D V+L settings, these tasks are generally dependent on the 3D features and text features extracted from the 3D settings and guidance text to adapt the downstream

tasks in a classic encoder-decoder manner. In early works [1, 6], 3D scene features are combined with text features through direct concatenation for downstream classifiers. Since attention mechanisms have proven to be successful in deep learning, many recent works [8, 15, 18, 42] have adopted transformer-based decoders as fusion module to improve performance and achieve better results.

*Scene Modification.* The field of scene modification has witnessed substantial progress in recent years. [44] uses GNN to construct the relationships between objects and their surroundings. Building on this, [38] introduces a method for inserting objects from CAD models into predicted positions based on the text prompt. Similarly, [28, 33, 34] utilized object selection and insertion techniques, simplifying the problem of scene generation to a selection of objects from the database and pose predictions for each object. However, a significant limitation of these methods is their heavy reliance on pre-generated point clouds or pre-stored CAD models. This dependence often results in inconsistencies with the surrounding environment and hampers seamless integration into scenes. Furthermore, this approach restricts the variety of objects that can be generated, contradicting the initial objective of accommodating open-ended text prompts. This constraint underscores the need for more flexible and adaptive techniques in scene modification, capable of generating a wider array of objects while ensuring harmonious integration with the existing environment. There are also some works [3, 14, 32] that are built based on neural radiance fields [25] that can synthesize indoor scenarios. However, these methods usually need images as input and the camera views of images are strictly restricted, which may not be feasible for certain tasks.

*Point Cloud Generation.* Many prior works have explored generative models over point clouds, including the use of autoencoder [2], flow-based generation [40], and generative adversarial neural networks (GAN) [16]. Besides, the Diffusion Model [12, 36], which has been proven to have great potential in the generative field, is widely applied. [23] treated point clouds as samples from a point distribution and reverse diffusion Markov chain to model the distribution of point. [43] introduce PVD, a diffusion model that generates point clouds directly instead of translating a latent vector to a shape. Yet, these studies did not demonstrate the capability to generate point clouds conditioned on open-ended text prompts. More recently, OpenAI introduced Point-E [26], a sophisticated model predicated on the concept of conditional diffusion, uniquely designed to generate point clouds directly, bypassing the need for latent vector translation. Point-E is also capable of producing colored point clouds in response to intricate text or image prompts, showcasing an impressive degree of generalization across a multitude of shape categories. Our object generation model is built upon the robust foundation provided by Point-E, capitalizing on its pre-trained model to enhance our system's capabilities.

## 3 METHODOLOGY

In this section, we introduce the proposed scene modification method. An overview of our model is presented in Fig. 2.

*Data Pipeline.* We transform the existing visual grounding dataset to accommodate instructed scene modification. As part of the data

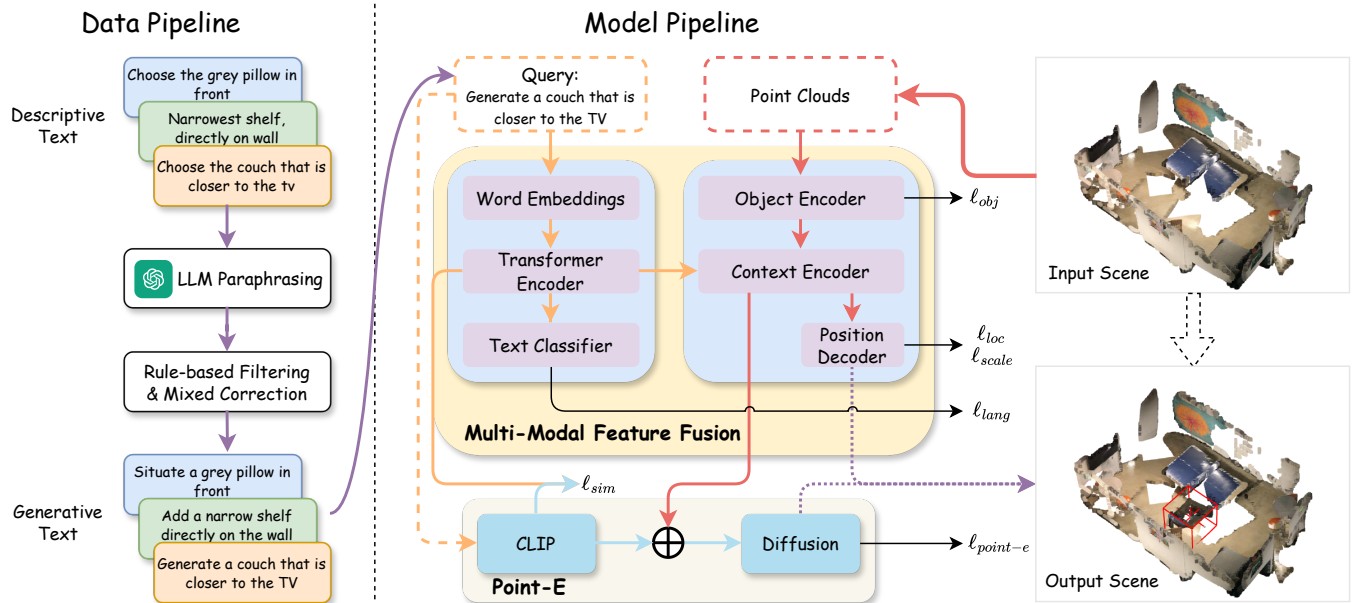

**Figure 2: Overview of our method. (a) A large language model (LLM) is used to paraphrase the descriptive text, combined with rule-based and manual corrections. (b) Upon receiving generative text as a query and point cloud input, our model integrates both object and language features to predict the final position. Besides, the language features are aligned across the model. The amalgamated features are then processed through the Point-E model to generate a realistic object.**

pipeline, descriptive texts are paraphrased by LLM to obtain generative instructions, which are then revised manually and by rules.

*Model Pipeline.* The scene modification process involves two stages: (a) locate the desired position using the grounding model; (b) create a new object based on the location and scene context using the text-to-point model. In the following sections, we will elaborate further on each module.

*Problem Statement.* The task of instructed scene modification involves generating a suitable target object $O_{tgt}$ within a specific scene $S$ based on a generative instruction. In our setup, a scene $S$ can be conceptualized as the ensemble of in-scene objects $\{O_{ctx,i}\}_{i=1}^{N}$. The spatial representation of object $O$ comprises its central location $l \in \mathbb{R}^3$, original size $s \in \mathbb{R}$, and normalized point cloud $\mathbf{p} \in [-1, 1]^{P \times C}$. For ease of understanding, we denote $S$ as:

$$\mathbf{L} \in \mathbb{R}^{N \times 3}, \mathbf{s} \in \mathbb{R}^N, \mathbf{P} \in [-1, 1]^{N \times P \times C} \quad (1)$$

where $N$ is the number of in-scene context objects and $C$ is the number of channels (e.g., $C = 6$ for XYZ-RGB points).

### 3.1 Dataset Transformation
To adapt the instructed scene modification task, our method transforms the ReferIt3D dataset [1] as shown in *data pipeline*. ReferIt3D dataset consists of 41K manually labeled (Nr3D dataset) and 114K machine-generated (Sr3D dataset) descriptions of specific targets in given scenes of the ScanNet dataset [10]. Each description entry illustrates the in-door location, type, and shape of the target object. Since the ReferIt3D dataset only contains descriptive texts, we leverage the GPT-3.5 [4] to paraphrase them into generative

instructions. The transformed datasets are noted as Nr3D-SA and Sr3D-SA, containing 155K generative instructions for 76 object classes, involving 1436 different scene scans of the ScanNet dataset.

Prompt engineering is used to facilitate the paraphrasing process. We construct well-designed prompting templates to instruct GPT-3.5 to perform paraphrasing. It should also be noted that human-labeled descriptions of Nr3D are generally more complex than those generated by machines of Sr3D. Even humans have difficulty distinguishing the correctly paraphrased ones from the incorrect ones in a *large corpus*. Therefore, we employ rule-based techniques to filter out the errors produced by GPT-3.5. The errors are then revised through an additional GPT-4 [27] round with manual corrections.

Detailed information regarding the prompt-based paraphrasing process, including the prompting templates and filtering rules, can be found in the Supplementary Material.

### 3.2 Multi-Modal Context Fusion
To accomplish multi-modal feature fusion, we decouple the fusion process into *feature extraction* and *cross-attention fusion*.

*Feature Extraction.* Point cloud features of all context objects are extracted by the object encoder. In practice, we use PointNeXt [30] rather than the commonly used PointNet++ [29]. For the language features of the query text, we adopt a Transformer Encoder-based language model (e.g., BERT [19]). Since the query text is relatively simple, only part of the encoder layers can handle language modeling. The object encoder and the text encoder produce the point

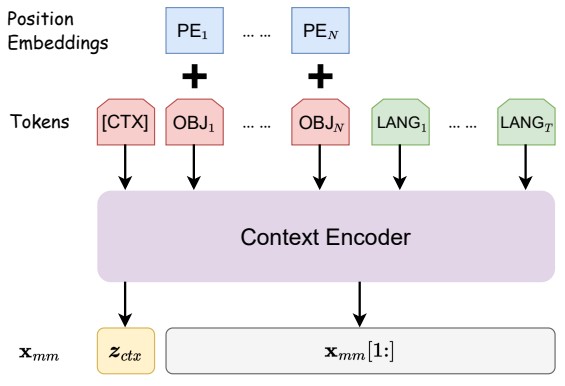

**Figure 3: Extraction of context vector $z_{ctx}$.**

cloud and textual features as $\mathbf{x}_{obj} \in \mathbb{R}^{N \times D}$ and $\mathbf{x}_{lang} \in \mathbb{R}^{T \times D}$, respectively. The dimension of the latent representation is $D$ whereas the token size of query text is $T$.

*Cross-attention Fusion.* The multi-modal features are fused by the object feature $\mathbf{x}_{obj}$ and the query feature $\mathbf{x}_{lang}$ using the cross-attention mechanism [37]. We adopt a standard Transformer Decoder as the context encoder. Prior to the cross-attention, a learnable token [CTX] is prepended to the front of object features as $\mathbf{x}_{ctx} \in \mathbb{R}^D$. Also, an additional object position embedding is applied to provide spatial information to the context encoder:

$$\text{PE}(\mathbf{L}, \mathbf{s}) = \text{LayerNorm}(\text{MLP}([\mathbf{L} \parallel \mathbf{s}])) \quad (2)$$

The multi-modal features are then calculated as:

$$\mathbf{x}_{mm} = \text{XAttn}([\mathbf{x}_{ctx} \parallel \mathbf{x}_{obj}] + \text{PE}(\mathbf{L}, \mathbf{s}), \mathbf{x}_{lang}) \quad (3)$$

where XAttn is cross-attention encoder, $\cdot \parallel \cdot$ is the concatenation operator, and MLP is the Multi-Layer Perceptrons.

The cross-attention mechanism integrates both the spatial feature of context objects and the query text feature. Alternatively, it can be considered a "scene encoder" that extracts the features from both the query text and the scene. As shown in Fig. 3, the context vector $z_{ctx}$ representing the entire scene and query is then extracted from the first token of $\mathbf{x}_{mm}$, corresponding to the position of the token [CTX].

### 3.3 Quantized Position Prediction

Given the inherent ambiguity and potential vagueness of many queries, predicting the location of objects poses a significant challenge for our model, as evidenced by our experimental results in Tab. 3, we introduce a technique known as *quantized position prediction*. This fundamental concept entails transforming a continuous coordinate system into discrete bins, simplifying the intricate regression problem into an easier classification task.

We divide the space into discrete bins and train the model to predict the normalized $xyz$ coordinates within each bin. The division procedure can be formulated as:

$$\tilde{l} = \left\lfloor \frac{l - \min_{xyz}}{\max_{xyz} - \min_{xyz}} \times B \right\rfloor \quad (4)$$

where $\tilde{l}$ is the normalized bin coordinate, $l$ is the original coordinate, $\lfloor \cdot \rfloor$ is the floor rounding function, $\max_{xyz}$ and $\min_{xyz}$ represent the maximum and minimum coordinate of each axis respectively, and $B$ is the total number of bins.

Furthermore, our practical experiments have revealed that objects within the same class often exhibit substantial variations in the $xy$-plane but tend to have similar $z$ coordinates. Thus, we separate the prediction process into two parts: one that addresses the $xy$-plane bin prediction and the other that addresses the $z$-axis bin prediction, and then concatenate them, formulated as follows:

$$\hat{l}_{xy} = \text{MLP}(z_{ctx}), \ \hat{l}_z = \text{MLP}(z_{ctx})$$
$$\hat{l} = [\hat{l}_{xy} \parallel \hat{l}_z] \quad (5)$$

where $\hat{l}$ is the predicted normalized coordinates. This normalized position is then restored to the original space's coordinates as the final predicted location.

### 3.4 Context-Aware Point Cloud Generation

We utilize the Point-E model [26] as our point cloud generation model. Point-E is a generative model developed by OpenAI for generating 3D point clouds from complex prompts based on Diffusion. We use the pre-trained model *base40M-textvec* provided by Point-E, which has been trained on ShapeNet [5]. Point-E's diffusion process, which is similar to other diffusion models, aims to sample from some normal distribution $q(\mathbf{x}_0)$ using a neural network approximation $p_\theta(\mathbf{x}_0)$.

In Point-E, *guidance* is used as a trade-off between sample diversity and fidelity in diffusion. Point-E inherits the classifier-free guidance from [13], where a conditional diffusion model is trained with the class label stochastically dropped and replaced with an additional $\varnothing$, using the drop probability 0.1. During the sampling, the model's output $\epsilon$ is linearly extrapolated away from the unconditional prediction towards the conditional prediction:

$$\hat{\epsilon}_{guided} = \epsilon_\theta(\mathbf{x}_t, \varnothing) + s \cdot (\epsilon_\theta(\mathbf{x}_t, \mathbf{y}) - \epsilon_\theta(\mathbf{x}_t, \varnothing)) \quad (6)$$

for guidance scale $s \geq 1$.

Several modifications are made to the Point-E model to better adapt it for context-aware generation tasks. One of the key changes involves the integration of context feature vectors $z_{ctx}$ with the text feature vectors $z_{CLIP}$ generated by the CLIP model [31] as shown in Fig. 2. Alignment between $z_{CLIP}$ and the text instruction feature vectors $z_{lang}$ produced by the transformer encoder is proposed to enhance cross-modal comprehension. This new combined feature vector is then used as input labels in the guided diffusion learning process, formulated as:

$$\ell_{sim} = \text{Cosine-Similarity}(z_{CLIP}, z_{lang})$$
$$\mathbf{y} = \text{MLP}(z_{ctx} \parallel z_{CLIP}) \quad (7)$$

The primary objective of this modification is to enable the diffusion model to effectively utilize contextual information from the scene and query text as *guidance*. This enables the modification of objects that are more seamlessly integrated with their environment.

### 3.5 Loss

The training process involves five losses, four for multi-modal feature fusion and one for Point-E diffusion, as illustrated in Fig. 2.

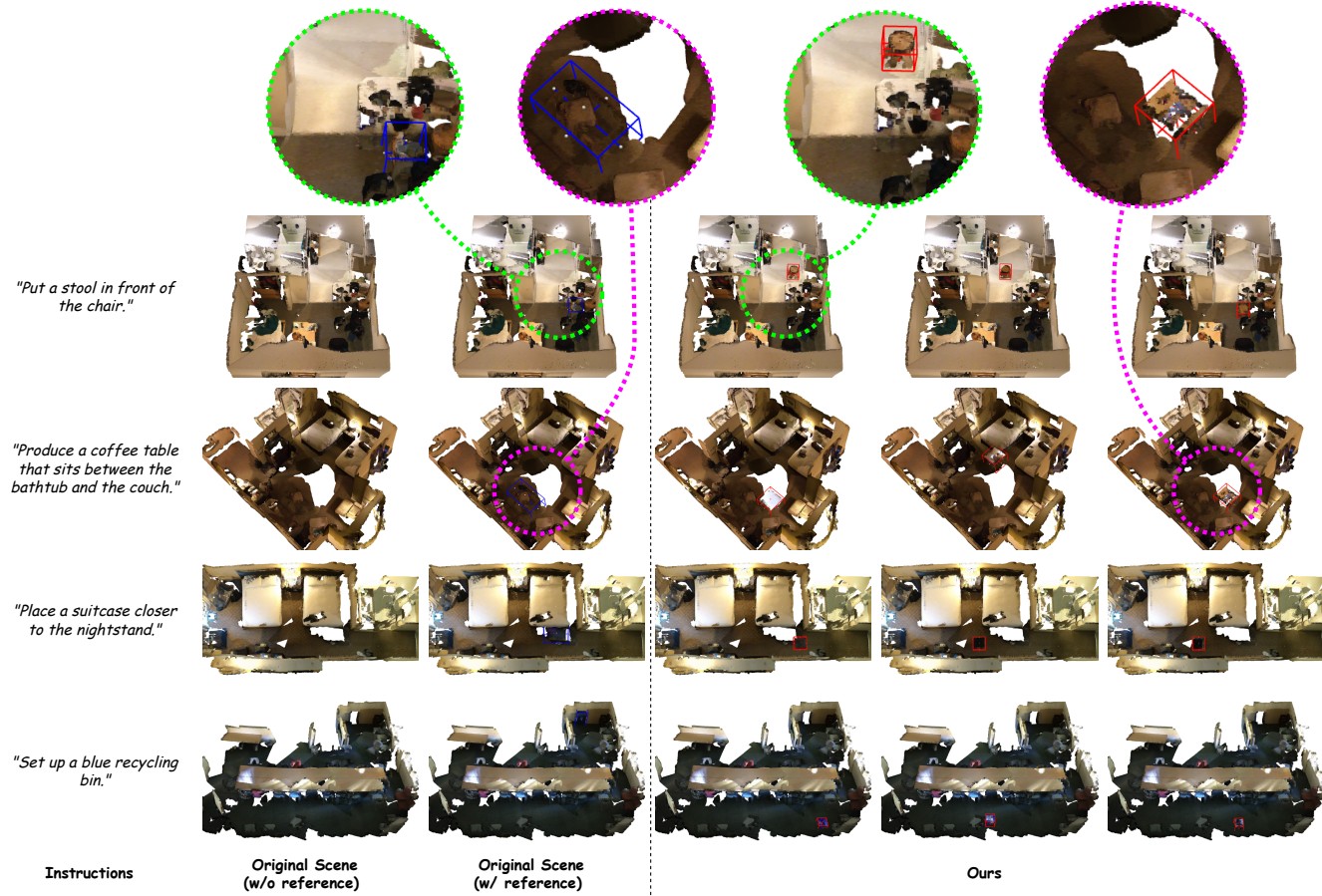

"Put a stool in front of the chair."

"Produce a coffee table that sits between the bathtub and the couch."

"Place a suitcase closer to the nightstand."

"Set up a blue recycling bin."

**Instructions**          **Original Scene (w/o reference)**          **Original Scene (w/ reference)**                    **Ours**

**Figure 4: Scenes before and after modification. Each row represents the scenes to be modified under different instructions. Different random seeds are used to generate the columns of the modified scene. Candidate locations are extracted from the Top-5 predictions. The bounding boxes of reference objects and generated objects are outlined in blue and red, respectively.**

Firstly, we have the loss $\ell_{obj}$, originating from the [15] and tailored for multi-modal feature fusion. Specifically, it is computed as the cross-entropy between the predicted object type of all context objects from the object feature $\mathbf{x}_{obj}$ and their ground truths. Next, we have the loss $\ell_{lang}$, which measures the discrepancy between the predicted type of the generated object from the query text feature $\mathbf{x}_{lang}$ and the ground truth. The third and fourth losses pertain to position prediction, represented by $\ell_{loc}$ and $\ell_{scale}$. These are supervised cross-entropy loss for target position prediction and L1 loss for object size, respectively. $\ell_{loc}$ is the combined loss of two MLPs defined in Eq. 5. Hence we could define the loss $\ell_{mm}$ as the total loss of multi-modal feature fusion:

$$\ell_{mm} = \alpha_{obj} \times \ell_{obj} + \alpha_{lang} \times \ell_{lang} + \ell_{loc} + \ell_{scale} \quad (8)$$

where $\alpha_{obj}$ and $\alpha_{lang}$ serve as weights for certain loss terms, both with the default value of 0.5. We also introduce $\ell_{sim}$ to align the multi-modal features extracted from the text encoder and CLIP model. Lastly, for point cloud generation supervision, we inherit the Mean Squared Error (MSE) loss from the Point-E model, denoted as $\ell_{point-e}$. We can therefore calculate the total loss as the sum of all losses during the training process:

$$\ell = \ell_{mm} + \ell_{point-e} + \ell_{sim} \quad (9)$$

## 4 EXPERIMENTS

### 4.1 Experimental Setup

*Dataset.* We train and evaluate our method on the Nr3D-SA and Sr3D-SA datasets generated in Sec. 3.1. For experiments trained on Nr3D-SA, only query data with explicit reference to the type of the target object is used, while all data is included in the Sr3D-SA settings. The datasets are divided into 80% for training and 20% for evaluation. The target object is separated from the other context objects in the scene and is set as the ground truth during training. In the training stage, we apply a 4-direction random rotation on the scenes. In the evaluation phase, the target object serves as a reference for assessing the generation quality.

*Implementation Details.* The dimension $D$ of latent representation throughout the model pipeline is set to 768. For the point cloud encoder backbone, we adopt the PointNeXt-L model based

on its state-of-the-art performance [30]. We adopt the pre-trained BERT$_{BASE}$ [19] with the first 3 layers as the query text encoder. The context encoder is a 4-layer Transformer Decoder for multi-modal feature fusion. The number of quantized bins is set to $B = 32$.

We implement our model in PyTorch and deploy point-cloud-based backbone models using the OpenPoints library [30]. Farthest Point Sampling (FPS) algorithm based on QuickFPS [11] is also used for efficient point sampling, with a setting of $P = 1024$ or $P = 2048$ as the point cloud size. For the training phase, our model is trained with a batch size of 16 for a total of 800,000 steps on both Nr3D-SA and Sr3D-SA datasets using ~480 RTX4090 GPU hours. We also train our model on only Nr3D-SA for quick verification, with a total of 320,000 steps. For optimization, we use AdamW with hyper-parameters $\beta = (0.95, 0.999)$, $\epsilon = 10^{-6}$ and weight decay of $10^{-3}$. The base learning rate for multi-modal fusion part and Point-E is $2 \times 10^{-4}$ and $4 \times 10^{-5}$ respectively, where the learning rate for BERT$_{BASE}$ and context encoder is set to $\frac{1}{10}$ of the base. Additionally, we employ a linear learning rate schedule from $2 \times 10^{-4}$ to $10^{-5}$.

## 4.2 Metrics

*Quality of Generation.* In the context of 3D generation evaluation, Earth Mover's Distance (EMD) [35] measures the similarity between two point clouds. Following previous works [2, 40, 43], we evaluate the quality of generated point clouds using the metrics of minimum matching distance (MMD), coverage (COV), 1-nearest neighbor accuracy (1-NNA), and Jensen-Shannon Divergence (JSD). A similarity between the distribution of generated and reference point clouds indicates a high degree of realism.

Given the fact that EMD and JSD are only capable of assessing the disparity in point distribution, thus merely providing an indirect evaluation of the generative performance, we propose an auxiliary metric to evaluate the quality of generated objects and the performance of language modeling. We employ a PointNeXt classifier trained on ReferIt3D to classify the generated objects. Furthermore, we observe that objects belonging to certain classes may have analogous shapes, such as *suitcases* and *boxes*. To mitigate this problem, we apply the Top-K estimation to classification accuracy, denoted as Acc@k. This approach allows us to mitigate the false negatives caused by similar shapes.

*Top-K Distance Estimation.* Due to the inherent ambiguity and vagueness in natural language, it is common to encounter multiple potential position matches for a single query. For example, when presented with a query like "place a chair in the corner" within a room with four corners, the model may produce four potential correct positions. However, only one corner is the true match according to the dataset. This array of potential matches adds complexity to accurately understanding and responding to user queries. To tackle this issue, we employ a technique called *Top-K Distance Estimation*, referred to as $\Delta l@k$.

This method allows the model to gauge its current performance more accurately by considering the Top-K closest match position, rather than relying on a single best match. By taking into account a range of closely matching responses, the model can better navigate the nuances and ambiguities of natural language and thus is less likely to be adversely affected by vague descriptions or queries.

**Table 1: Examination of the quality of the modified scene through visual grounding analysis. We utilized the MVT model [15], trained on the ReferIt3D dataset. Our modified scene was used as the test set, and we measured different *accuracy* across various difficulty levels, e.g., *Easy* and *Hard* mentioned in [15]. *Rnd.* means either the location or shape of the target object is randomly generated. *P.O.* stands for Point-E Only model and *GT* stands for ground truth.**

| Location | Shape | Easy(%, ↑) | Hard(%, ↑) | Overall(%, ↑) |
|----------|-------|-----------|-----------|---------------|
| Rnd. | Rnd. | 4.76 | 2.53 | 3.62 |
| Rnd. | P.O. | 13.58 | 6.78 | 10.11 |
| Rnd. | GT | *23.94* | *14.83* | *19.29* |
| Rnd. | *Ours* | **15.90** | **8.76** | **12.26** |
| GT | Rnd. | 14.41 | 7.78 | 11.03 |
| GT | P.O. | 36.71 | 25.64 | 31.07 |
| GT | GT | *61.44* | *47.28* | *54.22* |
| GT | *Ours* | **46.78** | **33.39** | **39.95** |
| *Ours* | Rnd. | 10.58 | 5.67 | 8.08 |
| *Ours* | P.O. | 35.06 | 24.00 | 29.42 |
| *Ours* | GT | *46.07* | *28.92* | *37.33* |
| *Ours* | *Ours* | **41.09** | **27.07** | **33.94** |

## 4.3 Experiment Results

*Visualization.* Figure 4 visualizes the qualitative results of our method on the evaluation of the Sr3D-SA dataset. The candidate locations of each object are chosen from the Top-5 predictions, while the point cloud is derived using different random seeds. Most generated point clouds are located close to the reference points and the shapes are consistent with the instructions. While some generated objects may vary from the references, they are oriented and sized following their surroundings.

Additionally, we notice that certain predicted locations diverge from the reference because of ambiguous instructions, such as the outcomes of *"Set up a blue recycling bin"*. To mitigate this ambiguity, positional prepositions (e.g., *"in front of the chair"*) can be employed to restrict potential locations to those proximate to the desired ones. Typically, the accuracy of location determination improves as instructions become less ambiguous.

*Diversity of Generations.* One of the key advantages of our approach to 3D object generation, which comes from the diffusion mechanism, is that diverse shapes can be generated for a given instruction, as shown in Fig. 5. This figure illustrates three distinct categories of point clouds generated from different random seeds. While maintaining consistency with the surrounding environment and instructions, our method creates meaningful variances in both shape and color. It allows the choice of the best shape to be made from a variety of options.

*Effectiveness of Instructions.* Moreover, since the shape of the generated object is determined by the instructions, the effectiveness of different instructions indicates the generalization ability of our approach. Fig. 6 provides results for the generated objects when different instructions are applied with slight variations. These results demonstrate that our approach can capture the differences

**Table 2: Snapshot of EMD values and classification accuracy for 32K objects generated from 32K randomly sampled generative texts from the test set, using our method and Point-E without feature fusion separately. Each class's proportion in the training set is also shown. MMD is multiplied by $10^2$ and JSD is multiplied by $10^1$.**

| Object Class | Ours | | | | | | Point-E Only | | | | | |
|---|---|---|---|---|---|---|---|---|---|---|---|---|
| | MMD↓ | COV↑ | 1-NNA↓ | JSD↓ | Acc@1↑ | Acc@5↑ | MMD↓ | COV↑ | 1-NNA↓ | JSD↓ | Acc@1↑ | Acc@5↑ |
| chair(7.58%) | 11.52 | 33.10 | 98.80 | 2.474 | 80.59 | 95.91 | 11.15 | 21.63 | 99.63 | 2.78 | 62.39 | 91.56 |
| door(6.72%) | 7.66 | 29.43 | 99.78 | 2.918 | 0.09 | 5.14 | 7.57 | 21.39 | 99.93 | 2.936 | 2.16 | 13.89 |
| trash can(4.78%) | 10.38 | 36.47 | 99.13 | 2.983 | 31.66 | 56.38 | 11.62 | 19.37 | 99.69 | 3.7 | 6.77 | 30.6 |
| window(4.76%) | 9.56 | 30.08 | 99.17 | 3.416 | 31.12 | 62.55 | 9.69 | 27.53 | 99.77 | 3.309 | 39.4 | 64.27 |
| table(4.70%) | 12.59 | 30.61 | 98.88 | 3.988 | 47.34 | 80.55 | 13.2 | 20.93 | 99.37 | 4.614 | 21.73 | 45.58 |
| cabinet(3.71%) | 12.22 | 36.38 | 98.57 | 2.960 | 16.95 | 64.00 | 10.82 | 27.64 | 99.5 | 2.974 | 9.21 | 38.84 |
| picture(3.53%) | 6.44 | 36.25 | 99.53 | 3.146 | 19.06 | 50.00 | 6.98 | 31.24 | 99.58 | 3.309 | 35.01 | 67.09 |
| shelf(3.42%) | 9.28 | 35.90 | 99.65 | 2.912 | 35.55 | 76.18 | 8.79 | 23.99 | 99.69 | 2.91 | 24.71 | 48.07 |
| lamp(3.17%) | 13.59 | 30.17 | 99.17 | 4.069 | 50.41 | 71.90 | 13.58 | 24.1 | 99.86 | 4.49 | 12.53 | 23.28 |
| desk(3.16%) | 14.59 | 35.11 | 99.56 | 3.660 | 11.56 | 44.67 | 13.96 | 22.85 | 99.85 | 3.839 | 1.9 | 15.33 |
| pillow(2.36%) | 10.96 | 40.94 | 98.82 | 3.033 | 51.18 | 69.69 | 10.71 | 37.24 | 99.24 | 3.18 | 26.58 | 50.23 |
| backpack(2.34%) | 12.02 | 36.79 | 97.65 | 2.758 | 33.07 | 66.34 | 11.54 | 27.52 | 99.21 | 2.414 | 27.94 | 68.31 |
| ⋮ | | | ⋮ | | | | | | ⋮ | | | |
| **Micro Avg.** | **12.05** | **34.08** | **98.89** | **3.634** | **30.75** | **56.45** | **11.50** | **26.87** | **99.40** | **3.620** | **18.37** | **39.85** |

**Table 3: Additive ablation study of sequentially applying different training techniques for Scene Modification tasks. Vanilla Transformer refers to the original transformer without any modification and directly predicts a coordinate, instead of a bin in quantized position prediction. Dash "-" is used as a placeholder for unavailable results. Acc@$k$ is the Top-$k$ classification accuracy of generated objects, whereas Acc$_{obj}$ and Acc$_{lang}$ are the accuracy of categorizing the context objects and instructions. $\Delta l@k$ stands for the minimum absolute difference between the predicted coordinate and the ground truth coordinate with Top-$k$ evaluation. $\Delta s$ measures the difference between the predicted and ground truth sizes. MMD is multiplied by $10^2$ and JSD is multiplied by $10^1$.**

| Ablate | Acc@1↑ | Acc@5↑ | $\Delta l@1$↓ | $\Delta l@5$↓ | MMD↓ | JSD↓ | $\Delta s$↓ | Acc$_{obj}$↑ | Acc$_{lang}$↑ |
|---|---|---|---|---|---|---|---|---|---|
| Transformer + Point-E | 18.96 | 39.34 | *2.179* | - | 13.29 | 3.575 | 0.161 | 51.95 | 89.63 |
| + FPS | 19.30 | 40.49 | *2.251* | - | 12.78 | 3.455 | 0.163 | 51.72 | 89.71 |
| Quantized Position ($B = 16$) | 18.93 | 39.11 | 2.599 | **1.264** | 12.50 | 3.538 | 0.163 | 51.70 | 89.97 |
| + Quantized Position ($B = 32$) | 23.10 | 44.23 | 2.654 | 1.302 | 12.37 | 3.471 | 0.162 | 52.07 | 89.88 |
| + $\ell_{sim}$ | 22.22 | 46.38 | 2.626 | 1.376 | 12.37 | **3.434** | **0.148** | 55.68 | 81.77 |
| + Sr3D-SA *(Ours)* | **30.75** | **56.46** | 2.486 | 1.379 | **12.05** | 3.634 | 0.209 | **63.47** | **92.31** |

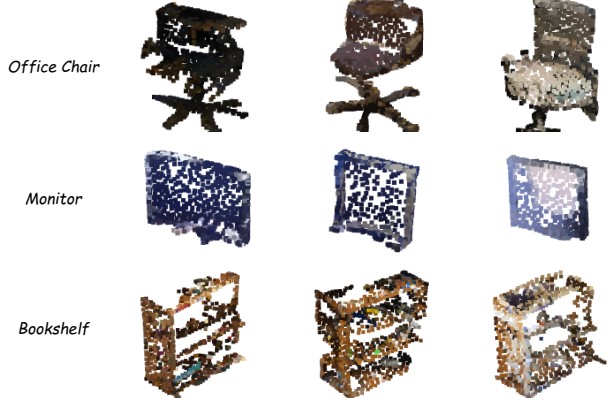

Office Chair

Monitor

Bookshelf

**Figure 5: Diversity. The leftmost column shows the category of the generated object to be generated from the instruction. Different generations under the same instruction are shown in each row.**

between instructions while maintaining the semantics of the target object. Generated objects can exhibit variations in color, shape, and location while remaining aligned with the provided instructions and the context of surrounding objects.

*Quantitative Result.* To assess the quality of the augmented scene, we employ the MVT model [15] to perform visual grounding task on three distinct scenes: randomly generated scenes, original ReferIt3D scenes, and our augmented scenes, as shown in Tab. 1. The goal of visual grounding is to identify the target object in a scene based on the text provided. There are no distractor objects of the same type as the target one in *Easy* tasks, whereas multiple objects of the same type are available in *Hard* tasks.

It is observable that our model is capable of generating scenes that are not only consistent but also easily recognizable by visual grounding models trained on the original dataset. Despite the inherent complexities involved in scene generation, which may lead to a certain degree of decline in overall visual grounding accuracy than the ground truth dataset, our model performs well in generating

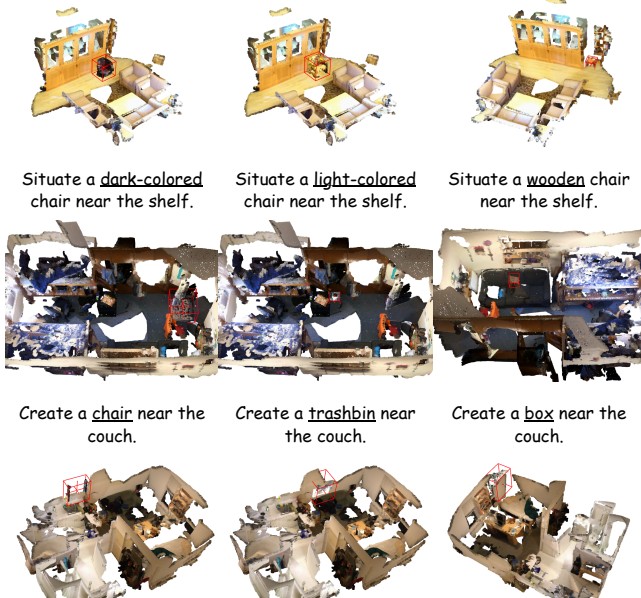

Situate a dark-colored chair near the shelf.

Situate a light-colored chair near the shelf.

Situate a wooden chair near the shelf.

Create a chair near the couch.

Create a trashbin near the couch.

Create a box near the couch.

Generate a window in the kitchen.

Generate a window above the sofa.

Generate a window beside the bed.

**Figure 6: Generated objects under instructions with slight variations. Each target object is created into the scene with a variation in *color, shape* or *location*. The generated object would be enclosed by a red bounding box.**

high-quality scenes. These generated scenes are more identifiable to the visual grounding model compared to those generated randomly or those generated by a single Point-E model with ground truth locations. Performance in *Hard* tasks also indicates the effectiveness of our approach in complex scenes.

Nevertheless, to conduct a comprehensive assessment of the performance of object generation, we sample 32,000 generative texts from the test set. For each generated object within this sample set, we compute the metrics in Sec. 4.2. We also perform experiments on a single Point-E model to compare the performance of our context-aware design, i.e., Eq. 7, as shown in Tab. 2. For a more comprehensive analysis of the object generation, we adapt the classification estimation with PointNeXt in Sec. 4.2 to categorize the generated objects, denoted as Acc@$k$ for Top-$k$ accuracy. The complete results are in the Appendix. It demonstrates that the ability of our model to generate objects is deemed reasonable.

*Ablation study.* In this section, we evaluate our method in different settings and strategies. We conduct an additive ablation study on the location and generation quality, as illustrated in Tab. 3. In the baseline model, only Nr3D-SA is used as training data, and the model is built on a bare backbone. Also, we evaluate the accuracy of identifying context objects and instructions with $\ell_{obj}$ and $\ell_{lang}$. It is noteworthy that the performance of $\Delta l$@1 degrades when the quantized position is applied. We observe, however, that the position predictor *without* quantized position exhibits significant

**Table 4: The results of utilizing the generated data as augmented data for visual grounding, serve as an illustration of downstream tasks. "w/" denotes the model trained with the combination of Nr3D dataset and the generated data, while "w/o" signifies the model trained solely with Nr3D dataset.**

| Metrics \ Dataset | Nr3D w/o Aug. | Nr3D w/ Aug. |
|---|---|---|
| **Easy**(%, ↑) | 35.2 | **42.5** |
| **Hard**(%, ↑) | 24.5 | **30.5** |
| **Overall**(%, ↑) | 29.7 | **36.4** |

under-fitting: most of the predicted locations remain close to the middle of the scene for a statistically minimal $\Delta l$, which contradicts the intended purpose. We hypothesize that directly predicting absolute position is more challenging as a regression task than predicting quantized position as a classification task.

## 5 DISCUSSION

Owing to the constraints of computational resources, we opted to sample 1024 or 2048 points. Nonetheless, for existing point cloud generation models [24], it is advisable to sample more points (e.g., 4096) to achieve a more real-life outcome. The overhead of generating point clouds is also considerably greater than that in prior works based on pre-built databases. Additionally, enhancing the generative model's capability may lead to further improvements in our model's performance. It is also important for our data-driven training process to expand the relatively limited dataset. Our results in the Appendix indicate that the performance is significantly lower for certain classes with less data.

## 6 APPLICATION

In addition to its use in AR and VR, our model has the potential for augmenting data in various downstream tasks, including visual grounding. To comprehensively explore our model's capabilities, we incorporate our generated data alongside Nr3D as the training set, employing MVT (referenced as [15]) as the visual grounding model. We then evaluate the performance with or without our generated objects as augmented data. Results indicate competitive performance compared to models trained solely on original Nr3D data. Further details are available in Tab. 4 and Appendix.

## 7 CONCLUSION

In this work, we present the first end-to-end multi-modal approach to generate augmented scenes conditioned on instruction. To obtain a proper dataset for scene augmentation, we use prompt engineering in conjunction with large language models to transform existing visual grounding data. Our method then utilizes both spatial and language features from the scene and instructions as guidance to the diffusion and locating processes. Furthermore, the experiment results exhibit the high capability of generating realistic objects at the appropriate locations according to various metrics and the visual grounding analysis. We hope this work will be a step towards the more practical applications of 3D human-computer interactions.

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
