# OpenReview forum: "Context-Aware Indoor Point Cloud Object Generation through User Instructions"
_acmmm.org/ACMMM/2024/Conference — MM2024 Poster_

### Official Review · Reviewer_bd5r · 2024-04-30

**Rating:** 4
**Confidence:** 3

**Summary:**

This work presents a novel end-to-end multi-modal deep network that generates 3D point cloud objects seamlessly integrated with surroundings based on textual instructions, addressing the limitations of traditional methods relying on pre-existing databases and enabling flexible and adaptive scene modification. To predict Top-K candidate positions, the paper proposes a quantized position prediction technique.

**Strengths:**

1. Previous works mainly focus on the transformation of a single object or generating point clouds from existing ones. Hence, this paper is theoretically well-motivated to generate new objects and incorporate them into scenes in an end-to-end manner.

2. The writing of the paper is clear and easy to follow, with few issues (detailed in the weaknesses section).

**Limitations:**

1. Did the author compare with the Nerf-based or Diffusion model-based method, which has been proven to have great potential in the generative field.

2. what benefits does the proposed dataset for scene modification tasks offer compared to previous datasets? Is the scale of the proposed data too small, and can it meet the future needs of this task? Given the relatively limited amount of training data, the performance in generating certain object categories is significantly inferior. Has the author considered presenting any failure cases to illustrate these shortcomings?

3. Could equation 8 list the formulas for each loss in detail for easier understanding?

4. In the proposed framework, how does the learnable token [CTX] learn contextual information?

5. The Point-E generation model used still has room for improvement. Has the author compared the effects of other newer point cloud generation methods?

**Suitability:**

3

---

### Official Review · Reviewer_hrhN · 2024-05-22

**Rating:** 2
**Confidence:** 3

**Summary:**

This paper presents a novel end-to-end multi-modal approach to generate augmented point-cloud scenes conditioned on textual instructions. It utilizes both spatial and language features as guidance to the diffusion and locating process. Experimental results show its strong capability of generating realistic objects at the appropriate locations according to various metrics.

**Strengths:**

(1)The purpose of this article is to generate new point clouds based on text information in existing point cloud spaces. This task has a certain degree of novelty and can have a certain impact on AR and VR technology.
(2)The article uses a large language model to process the existing visual grounding dataset to get generative instructions, which is helpful for generating text content.
(3)The fusion of spatial and textual features is considered in the process of point cloud generation, which can ensure semantic and spatial consistency.
(4)The author of the paper has already made some of the code open-source, which helps readers understand the content of the work.

**Limitations:**

(1)The article's description of the model pipeline section is not clear enough, and it does not follow the diagram in Figure 2 to unfold the description in sequence. The gap between the text and the diagram is relatively large, making it difficult for readers to understand.
(2)The article does not provide a good explanation of the loss functions in each module, which looks like a stack of multiple modules without seeing the relationships between them.
(3)In the process of point cloud generation, in addition to considering semantic features, the influence of lighting uniformity, surface texture, and other factors should also be considered, which were not considered in the paper.
(4)The experimental part of the article only compared the results with Point-E, which seems relatively ordinary.
(5)The content of the article is very attractive, but overall, the author's preparation in writing and methods seems to be insufficient, and they did not provide a good expression.

**Suitability:**

3

---

### Official Review · Reviewer_nv2C · 2024-05-22

**Rating:** 5
**Confidence:** 3

**Summary:**

This paper presents a novel in-context point cloud generation system that can be applied to scene editing, AR/VR, and various applications. The proposed pipeline features several key technical contributions, including data paraphrasing, multi-modal context fusion, quantized position prediction, etc. This paper has discussed several key factors in training the multi-modal feature fusion module. Through comprehensive qualitative and quantitative evaluation, this paper demonstrated the effectiveness of the proposed method.

**Strengths:**

- This paper presented an in-depth analysis of multi-modal feature fusion module training, including the trainable model and loss designs.
- The proposed methods address the spatial location grounding, which is a key problem in text-guided 3D understanding and generation.
- The proposed method is evaluated through several benchmark tests with promising results.

**Limitations:**

- Although we recognize the technical contribution of this paper, it is hard to ignore there are quite a few overclaims in the paper writing. For example, "Our model revolutionizes scene modification (Page 1, Line 16)". It is hard to be convinced that "revolutionizes" is the right word to use.
- The proposed model pipeline does not take into account the scene viewport directions and positions. In a real-world application, end users' scene observation will likely be impacted by the viewport they are looking at. For example, objects can be occluded from certain perspectives. Including camera viewport information in the context can also potentially improve the models spatial understanding ability.

**Suitability:**

3

---

### Meta-Review · Area_Chair_DLoi · 2024-07-01

**Recommendation:** Accept (Poster)
**Confidence:** 4

**Metareview:**

The paper presents end-to-end multi-modal diffusion-based deep neural network model for generating point-cloud scenes conditioned on textual instructions.

The paper received only 3 reviews but all the reviewers showed to be confident in the topic. Even if all the reviewers agreed on the technical contribution and relevance of the paper for ACM Multimedia community, there are some concerns on the presentations (e.g., lack of clarity in the description of the proposed pipeline, loss functions and equations). Some limitations on the experimental comparisons have been also raised by reviewers.


The paper seems to be a good fit to be presented as a poster at ACM Multimedia. Please, in the camera ready address reviewers' comments clarifying better the description of the proposed framework (included selected loss function and equations), and clarify better the contributions of the presented datasets ("it can be easily used with other visual grounding datasets, allowing for the generation of additional data iteratively" some specific examples can be added). I would like also to mention that most of references are from computer vision and machine learning communities such as CVPR and NeurIPS, the paper can be also shaped to better fit as a publication of ACM Multimedia.